# The Impact of Early Saliva Interaction on Dental Implants and Biomaterials for Oral Regeneration: An Overview

**DOI:** 10.3390/ijms23042024

**Published:** 2022-02-11

**Authors:** Marcel Ferreira Kunrath, Christer Dahlin

**Affiliations:** 1Department of Biomaterials, Institute of Clinical Sciences, Sahlgrenska Academy, University of Gothenburg, P.O. Box 412, SE 405 30 Goteborg, Sweden; christer.dahlin@biomaterials.gu.se; 2Department of Dentistry, School of Health and Life Sciences, Pontifical Catholic University of Rio Grande do Sul (PUCRS), P.O. Box 6681, Porto Alegre 90619-900, RS, Brazil

**Keywords:** saliva, oral regeneration, interaction, biomaterials, bone regeneration

## Abstract

The presence of saliva in the oral environment is relevant for several essential health processes. However, the noncontrolled early saliva interaction with biomaterials manufactured for oral rehabilitation may generate alterations in the superficial properties causing negative biological outcomes. Therefore, the present review aimed to provide a compilation of all possible physical–chemical–biological changes caused by the early saliva interaction in dental implants and materials for oral regeneration. Dental implants, bone substitutes and membranes in dentistry possess different properties focused on improving the healing process when in contact with oral tissues. The early saliva interaction was shown to impair some positive features present in biomaterials related to quick cellular adhesion and proliferation, such as surface hydrophilicity, cellular viability and antibacterial properties. Moreover, biomaterials that interacted with contaminated saliva containing specific bacteria demonstrated favorable conditions for increased bacterial metabolism. Additionally, the quantity of investigations associating biomaterials with early saliva interaction is still scarce in the current literature and requires clarification to prevent clinical failures. Therefore, clinically, controlling saliva exposure to sites involving the application of biomaterials must be prioritized in order to reduce impairment in important biomaterial properties developed for rapid healing.

## 1. Introduction

Saliva present in the oral cavity is recognized as the first liquid that interacts with materials or compounds coming from the external environment [1,2]. Furthermore, its presence is crucial in several basic food processes, such as lubrication, digestion and solubilization in the oral environment [2,3]. Saliva is predominantly water (99%); however, there is a high range of proteins, different minerals, dead cells and a large amount of bacteria associated with the composition [1,4]. The interaction of saliva and its compounds with biomaterials applied in oral rehabilitation or tissue regeneration is inherent, thus impacting the creation of small salivary pellicles [5,6] that may cause some chemical–physical–biological alterations in biomaterials.

Currently, biomaterials developed for oral rehabilitation and tissue regeneration, such as dental implants, membranes and bone substitutes, are made with extreme detailing of properties aimed at the best biological responses when interacting with oral tissues [7,8,9]. Alterations such as the micro- and nanotexturization of surfaces are developed with the aim of accelerated cellular response and high adhesion in contact with bone cells [7,10]; the porosity and mechanical strength are controlled in membranes and in bone substitutes for better tissue responses in guided bone regeneration [8,11,12]; and wetting properties are altered for superior physical–chemical interactions within surfaces and cells in oral tissues [13,14]. However, the early interaction with saliva can cause significant changes in these properties that can modify the desired effectiveness in biological terms.

A recent report showed the loss of hydrophilicity on dental implant surfaces after interaction with human saliva in vitro for 10 min, showing a negative outcome caused by the early interaction with saliva [14]. Additionally, surfaces with microroughness and high hydrophilicity showed greater adhesion of saliva proteins compared to smooth and less hydrophilic surfaces, suggesting that surface properties influence the interaction with saliva [15]. On the other hand, saliva shows some positive features against bacterial colonization due to its lubrication and viscosity properties; therefore, some investigations demonstrated greater bacterial colonization in static biomaterials without the presence of saliva compared with biomaterials embedded in saliva [16,17]. However, as a counterpoint against the beneficial characteristics, there are many reports showing the growth of bacterial biofilms in biomaterials applied in tissue regeneration after interaction with saliva, compromising properties and biocompatibility factors, normally caused by dysbiosis or imbalance in the superficial region, and generating an enrichment of substances suitable for pathogens [18,19,20,21].

It is well known that saliva has some influence on all biomaterials inserted in the oral environment, especially on those that are subjected to exposure for long periods of time such as dental implants and components for prosthetic rehabilitations [16,22]. However, the early influence of salivary interactions with biomaterials for oral regeneration remains poorly understood due to numerous innovative modifications in biomaterials and due to the difficulty of analyzing biomaterials directly after their application in the oral region. The aim of this review is to highlight the main early alterations caused in the physicochemical properties of biomaterials for oral regeneration exposed to saliva and to demonstrate their possible alterations in terms of biocompatibility and contamination when saliva interactions occur. The review will conclude with clinical considerations about possible effects derived from early salivary interaction with biomaterials for oral regeneration in the clinical environment.

## 2. Search Strategy

An electronic search in the PubMed Medline, Scopus and Google Scholar databases was conducted to identify in vitro, in vivo and clinical studies assessing the interaction of saliva with biomaterials for oral regeneration. The electronic search was carried out using key words and MeSH terms: “saliva” or “saliva contact” or “saliva interaction” or “saliva proteins” and “biomaterials” or “implants” or “dental implants” or “membranes” or “bone substitutes” or “bone regeneration” or “cells” or “bacteria”. The inclusion criteria for this study were as follows: (1) English written studies, (2) reviews, (3) meta-analyses, (4) clinical trials, (5) animal studies and (6) in vitro studies. The titles and abstracts were evaluated individually to find possible relevant studies for this review. The key studies (85) were then selected independently and analyzed to summarize the possible effects caused by the saliva interaction with biomaterials for bone regeneration.

## 3. Saliva Composition

Saliva is mainly secreted by the major glands (parotid, sublingual and submandibular) and minor glands such as labial, palatal and buccal glands [1,23]. The interaction of saliva with any substrate inserted in the oral environment results in the formation of a thin film with thickness measured in nanometers up to 1 h and in micrometers (approximately 1.3 μm) after 24 h [1,5].

The essential component of saliva is water (99%); however, inorganic elements such as NaCl, KCl, NHCO_3_, HPO_4_ and CaCO_3_ and inorganic constituents such as proteins, mucin, serum albumin and globulin, enzymes, epithelial cells and lymphocytes are found [4,23]. In recent years, more than 1400 salivary proteins have been identified [24] and have specific actions in oral matter, such as lubrication, digestion, solubilization, defense, and remineralization. The most commonly identified proteins in saliva are shown in Table 1.

The absorption of proteins from saliva on certain substrates depends mainly on noncovalent interactions [1]. However, biomaterials used in oral regeneration, such as dental implants, membranes and bone substitutes, have different surface characteristics that aim for greater cellular interactions, such as hydrophilicity, porosity changes, greater roughness and high surface energy [25,26,27]. These superficial alterations can modify the interactions with saliva proteins, and the manufactured characteristics for better biological performance in these biomaterials can be affected after interaction with saliva (Figure 1).

## 4. Effects of Saliva Interaction on the Physical–Chemical Properties of Biomaterials for Oral Regeneration

### 4.1. Biomaterial Morphology/Roughness

Morphological characteristics and roughness measurements are essential features for biomaterials with rapid healing properties, with surface modifications being particularly important for dental implants and topographies for membranes [28,29]. Different morphologies have been proposed, aiming for immediate interactions at the micro- and nanoscale level with the extracellular membrane of cells [10,27,28]. Moreover, differences in roughness parameters have been reported to act on the adhesion properties of cells, as well as in the initial contact [30,31]. As described previously, saliva has the potential to create a thin layer over the substrate in a short period of time. In some cases, where specific nanotopographies or nanoroughness measurements are developed for biomaterials, this layer may affect or cause small alterations in these properties.

Normally, studies applying membranes for bone regeneration focus more on the investigation of biodegradation characteristics rather than early contact with saliva. After only 4 weeks of artificial saliva exposure, Liao et al. [32] observed some morphological differences or biodegradation in their specific carbonized hydroxyapatite/collagen/PLGA composite membrane. However, the literature is very scarce about the initial details that may cause possible changes in morphologies induced by the formation of a salivary pellicle. The current findings demonstrated only some adherence of impurities in dental implant surfaces after exposure for 10 min [14], thus suggesting a promising topic to be further investigated in dental biomaterials for oral rehabilitation.

Regarding roughness properties, some considerations are very clear within the studies done today. Substrates with high roughness parameters showed increased adhesion of saliva proteins compared with smooth surfaces; this statement is corroborated by diverse studies with different standards of roughness and different base materials applied [14,15,33]. Furthermore, specific saliva proteins, such as mucin glycoprotein 2 and lactoferrin, were reported as high-affinity proteins to rougher dental implant surfaces [33,34]. Additionally, Souza et al. [35] demonstrated changes in the proteomic profile of saliva protein adsorption to SLA (sand-blasted/acid-attacked) surfaces compared with machined surfaces. In contrast, the authors reported that roughness is not a parameter that creates specificity of adhered proteins because most proteins found on surfaces are similar, varying only in the quantification [15].

### 4.2. Biomaterial Surface Composition

The interaction with saliva may cause some specific adherence of different atomic elements present in saliva. Saliva in the natural state is composed of a large quantity of additional compounds present in the mouth. In addition to saliva proteins, minerals such as calcium and phosphate are easily found in saliva, ions from metals might be found in persons that have metallic devices in the oral environment and a wide number of impurities may be exhibited in the mouth due to the continuous process of feeding and digestion.

The discussion about early contact with saliva with biomaterials for oral rehabilitation and the possible atomic alterations caused by the interaction is another topic with few reports in the literature. Kunrath et al. [14] applied dental surfaces (machined, rougher and nanotexturized) to human saliva exposure for 10 min and 1 h. The results showed nonsignificant findings regarding the composition of the surfaces, only demonstrating that few impurities and minerals had adhered to the surfaces [14]. Moreover, other authors reported the stability of electrochemical and corrosion behavior after exposure (7 days) in artificial saliva of different commercial implant surfaces [36]. In contrast, another study demonstrated the influence of saliva pH on corrosion resistance using pure titanium and alloys for dental implants [37]. Lower pH values in saliva may represent an increased corrosion rate and kinetics [37,38].

For bone substitutes, only scarce information is available revealing a nonalteration in the atomic stability of granules derived from pork bone sludge after exposure for 60 days in artificial saliva. The authors recommended further studies involving contaminated saliva to verify the differences with a real environment [39]. On the other hand, De Aza et al. [40] showed a material structural transformation (pseudowollastonite–α-tricalcium phosphate bioeutectic) internally and superficially after 30 days of exposure in human parotid saliva, creating a hydroxyapatite-like phase in the related bone substitute.

Studies reporting analyses in short periods of time were not found in the literature, and the understanding of the possible changes in biomaterials when in early interaction clinically with human saliva remains poorly investigated regarding the structure and surface composition.

### 4.3. Biomaterial Wettability

There is a consensus in the literature that hydrophobic solid surfaces are more attractive to saliva and their proteins [1,41,42]. On hydrophobic surfaces, saliva proteins adhere and group to the surface due to the process of not spreading under the entire substrate, thus managing to create a film. However, on hydrophilic surfaces, this process becomes more difficult due to the spacing and spreading of proteins when in contact with the surface with high surface energy [41,42]. Investigations have shown that hydrophilic surfaces provide the adhesion of specific saliva proteins, such as aPRP, bPRP, cystatin S and Statherin, but in less variety and quantity than hydrophobic surfaces [41,43]. Moreover, the interactions between saliva/substrate do not behave similarly to bacterial biofilm growth. The formation of the saliva pellicle is explained by the adhesion of a molecular film composed predominantly of saliva proteins [44]; thus, after the protein superposition over the entire surface, the chemical intercommunication between substrate/proteins decreases. Additionally, the saliva film creates properties of lubrication and viscosity derived mainly from the mucin protein, which complicate the film development at a higher thickness [44].

Schweikl et al. [45] reported the influence of wettability characteristics in applying different base materials used in dentistry, such as titanium, PTFE, PE and PMMA, regarding the adsorption of human saliva proteins. The authors showed slightly higher adsorption of saliva proteins to hydrophobic surfaces; moreover, most material surfaces with hydrophobic characteristics after saliva interaction had decreased measurements of surface angle contact due to the formation of the saliva pellicle [44]. Similarly, other studies found significantly higher adhesion of saliva proteins to hydrophobic surfaces on solid surfaces (Figure 2) [42,46].

Currently, there are huge numbers of publications referring to the advantages of hydrophilic surfaces for dental implants or for bone regeneration materials [48,49]. Hydrophilic surfaces provide improvement in some important behavior aspects for bone cells and soft tissue cells [48,49,50]; therefore, companies are developing biomaterials with superhydrophilic properties to achieve the best response possible when inserted in the desired oral environment. However, reports demonstrated that the early interaction of hydrophilic dental implant surfaces with human saliva showed the loss of this characteristic after interaction [14,47]. Additionally, Muller et al. [46] reported similar results that the saliva pellicle could change the hydrophobic/hydrophilic properties and vice versa.

## 5. Effects of Saliva Interaction on Biocompatibility Properties of Biomaterials

### 5.1. Biocompatibility

#### 5.1.1. Dental Implants

Dental implants are usually placed with extreme accuracy associated with a free-contaminated environment during insertion in a planned site. However, controlling salivary production around the surgical site may be a complicated procedure. Normally, during implant insertion, the implant surface touches the bone tissue and/or the soft tissues that interact with saliva present in the mouth. Moreover, if the surgery is not assisted by surgical aspiration, saliva may touch the biomaterial due to normal saliva excretion by the major and minor salivary glands. To minimize the risk of contamination, authors reported techniques suggesting the use of a rubber dam during the placement of dental implants and sinus lift procedures in order to prevent saliva contamination in surgical sites [51].

The first interactions with oral tissues occur when the implant is inserted and are associated with blood protein adherence, pro-osteogenic cell adhesion and bone cell attachment; therefore, dental implant surfaces are designed to promote a beneficial response and attachment for these specific cells [7,10]. However, there are a huge number of reports showing the impairment of cell viability after saliva interaction with surfaces compared to surfaces without interaction [14,21,47,52]. The authors demonstrated that saliva interaction with biomaterials prior to cell culture impaired the behavior of the MC3T3-E1 osteoblast cell line [14], MG63 human osteoblasts [52] and bone marrow cells derived from Sprague-Dawley rats [21] (Figure 3); additionally, more reports showed the same problematic behavior for soft tissue cells such as human gingival fibroblasts (HGFs) [47,53]. On the other hand, Sun et al. [54] revealed that the addition of an isolated salivary protein (histatin-1) to a titanium surface improved the spread and some features of MC3T3-E1 osteoblast cells after culture for some days. Moreover, Caballe-Serrano et al. [55] demonstrated, using in vitro models, that bone tissue contaminated with saliva showed less osteoclast reabsorption and presented some differences in the immune response, which highlights the need for further investigation regarding the effects of saliva interaction in clinical or animal models.

Similarly, a study using animal models and implants contaminated with saliva prior to insertion showed osseointegration parameters with reduced values in comparison to implants not exposed to saliva, for example, lower bone-interface contact (BIC) [56]. Additionally, this specific report applied highly contaminated saliva collected from a patient with periodontitis, and the results could not determine whether saliva alone was responsible for the decreased results in terms of osseointegration. However, the study simulates a clinical reality that is the presence of high-contaminated saliva and the application of biomaterials in unhealthy patients.

#### 5.1.2. Membranes and Bone Substitutes

As described in the previous chapter, the interaction with saliva has been studied with more emphasis in dental implants regarding the effects caused in the biological responses after saliva contamination. In the same way, a similar interaction may occur with biomaterials for guided bone regeneration, such as membranes, particulate bone substitutes, grafts and scaffolds. The clinical application of these biomaterials is normally aimed at the same sites where implants are placed, and the interaction with saliva is difficult to control.

Nevertheless, the literature is scarce regarding the early interaction with saliva in biomaterials for guided bone regeneration. Previous studies mostly employed cell cultures applying saliva to the substrate without any additional biomaterials, suggesting negative results for all the different types of cells applied, such as osteoblast-like cells or fibroblasts that interacted with the saliva [57,58,59].

Thus, consistent research about the early reactions of biomaterials for guided regeneration is still necessary to provide knowledge about the integrity and biocompatibility after possible saliva interactions in these biomaterials. Table 2 summarizes studies involving different cells and their reactions to contaminated substrates with saliva, suggesting in 100% of the studies that saliva impairs the behavior or proliferation of specific cells. The opposite is only revealed when studies propose the isolation of specific proteins (histatin-1) to adhere to the substrate and create a beneficial effect for the cells [54,60].

### 5.2. The Role of Saliva and Bacterial Contamination

#### 5.2.1. Dental Implants

Biomaterial contamination in the initial stage of healing is one of the major causes of clinical failure and early loss of dental procedures [62,63]. The mouth is a human body area with an expressive quantity of bacteria, and saliva becomes an intraoral conductor for these different bacteria [64]. Associated with these characteristics, numerous unhealthy oral conditions, such as periodontitis, gingivitis, fungi and infections in adjacent teeth, may stimulate proliferation and the presence of more types of bacteria in contact with oral saliva.

Following these statements, dental implants inserted in contaminated patients with oral diseases or adjacent infections interacting with local saliva present a high risk of surface contamination. In addition, the conduction of bacterial cells to the bone tissue or to the intragingival tissue can occur from contamination with saliva. Surfaces developed for dental implants contaminated with saliva were shown to stimulate the virulence of *Candida albicans* [65]; in addition, the salivary protein mucin was demonstrated to be a receptor for adhesion of *C. albicans* [66].

Other studies revealed that the salivary film formed over the surface could develop a favorable environment for bacterial nutrition and metabolism on the surface of implants [1,67] or teeth [1,68] (Figure 4). Rough surfaces provided greater adhesion of cells and proteins from saliva and likewise showed greater adhesion of the *S. oralis* bacteria to surfaces made with microscale roughness [14,67,69]. Surface physicochemical properties such as morphology, roughness and wettability can significantly alter the level of bacterial adhesion associated with saliva, suggesting that each substrate may differently influence the adhesion of contaminated saliva [70,71].

Promising features for dental implant surfaces have been reported to control bacterial proliferation with the application of nanotexturizations or antibiotic-loaded surfaces [72,73], suggesting a protective coating for contaminated environments. However, preclinical and clinical studies are rare regarding assays with saliva-contaminated implants associated with innovative surface treatments. Jinno et al. [56] demonstrated the insertion of implants contaminated with human saliva derived from one patient with confirmed periodontitis; the results showed impairment in the osseointegration process for all contaminated implants compared to noncontaminated implants. Clinical studies revealed the presence of bacterial contamination that may promote peri-implant resorption around the implant–abutment interface, the region exposed to saliva, suggesting the requirement of special treatment (Chlorhexidine 0.20%) to improve the tissues response [74,75]. Therefore, the critical issue about the use of biomaterials close to contaminated saliva in oral sites should be confirmed with long-term clinical studies.

#### 5.2.2. Membranes and Bone Substitutes

Equivalent results were shown for biomaterials manufactured for oral regeneration, such as modified membranes or hydroxyapatite substitutes [76,77,78,79]. The salivary film deposited over the biomaterial surface can mediate the adhesion of important bacteria present in oral infections [68,78]. Moreover, organic biomaterials may promote a favorable environment for bacterial metabolism due to the degradation of specific molecules compared to inorganic biomaterials or synthetic biomaterials [76]. On the other hand, Lee et al. [79] revealed, using in vitro models, that their PMMA membranes possess antibacterial properties even after contamination with saliva. Studies applying saliva-contaminated biomaterials are scarce in the literature due to the associated negative effects, mainly in animal models and clinical models where the tissues can be damaged with these bacteria. Thus, it remains unclear how early saliva contact may impair healing around these biomaterials. Table 3 summarizes the main studies using biomaterials for bone regeneration with saliva and bacterial interactions.

## 6. Limitations of Studies Applying Saliva-Contaminated Biomaterials

In vitro or in vivo studies involving saliva may have some limitations when compared with clinical investigations. The first issue about laboratory studies with saliva is related to the use of human saliva or artificial saliva. The comparison between human and artificial saliva cannot be translated with complete accuracy; although the solution composition is almost the same, the presence of contaminants, impurities or cells cannot be created with precision in artificial saliva, as well as some exclusive properties of human saliva [44,83].

A second and determinant problem reported when applying human saliva in laboratory studies appears to be the methodology chosen to collect saliva from humans. Studies have applied different techniques to collect human saliva, such as collecting directly from the human mouth without storage or treatment, using techniques for saliva stimulation, filtering the saliva before application in the samples and application of different methodologies for storage and later investigation [14,15,35,56,84]. These different techniques need to be carefully understood to be able to compare the results with the different studies performed. Moreover, the clinical translation of some results applying different techniques for saliva collection requires attention by the readers due to the possibility of masking important saliva characteristics when compared to clinical human saliva, which may prevent some clinical conclusions.

From our point of view, the appropriate methodology to be applied in studies using saliva is focused on the search for maximum similarities to clinical human saliva, including all contaminants and impurities, and, if possible, should be associated with a direct application in the experiment without storage. On the other hand, several studies have demonstrated the stability of saliva composition and components using processes to store saliva at cold temperatures [14,84], and this alternative should be considered to develop and facilitate laboratory studies with scientific relevance.

## 7. Clinical Significance

The interaction between saliva and biomaterials is extremely difficult to control in the clinical setting. Normally, surgical procedures require “four hands” handling or more “hands” to create totally free-saliva environments applying good aspiration and perfect management. Restorative treatments are commonly employed with absolute isolation to prevent saliva contamination. In addition, the use of latex gloves by clinicians is often a factor for saliva stimulation in some patients.

Saliva is present on all tissues exposed to the oral environment, including blood when free in the mouth, soft tissues when accessed for gingival treatments and bone tissues when exposed for surgeries (Figure 5) [85]. Therefore, a minimal interaction with saliva is almost inherent. The formation of salivary pellicles over biomaterial substrates is quick and may change some specific properties, as mentioned in this review. Additionally, with intense biomaterial exposure to saliva, the positive biological response can be affected, reducing important cellular reactions [53,61]. In addition, saliva contamination can promote an environment for high bacterial metabolism and possible proliferation [68,78].

Thus, after removing the limitations of laboratory studies, clinical studies showed that saliva interaction must be controlled with the maximum of effort to prevent alterations in all types of biomaterials previously inserted in the oral environment. The early saliva interaction is not a clear factor that results in unsuccessful treatments; however, it is reported as a significant element that may impair positive properties developed for rapid healing in biomaterials applied in oral regeneration [14,21].

Additionally, this review has some limitations due to the compilation of results from different authors providing a wide visualization of all possible alterations caused by early saliva interaction investigated until the current moment, in addition, due to the narrative methodology. Therefore, studies applying preclinical and clinical investigations should be prioritized for clinical conclusions.

## 8. Conclusions

Within the limitations of this critical review, different issues about the interaction between saliva and biomaterials for oral rehabilitation were clearly identified and elucidated. A constant tissue interaction and minimal biomaterial contact with saliva are almost inherent in surgical procedures involving dental implants, bone substitutes and membranes for guided regeneration. Therefore, some conclusions can be made after critical analysis of the investigations explored in this study:Salivary pellicle formation over biomaterials is an extremely quick and natural process that occurs within the first minute of interaction with saliva. The pellicle thickness depends on the exposure time to saliva and on the physical–chemical properties of the substrate.Accordingly to the physical–chemical studies explored, hydrophilic and hydrophobic characteristics are clearly altered by the interaction with saliva, causing substantial changes in biomaterials with surfaces designed for rapid healing. Moreover, rougher biomaterial surfaces showed high salivary protein adsorption.Accordingly to the basic biological studies analyzed, biomaterial biocompatibility with different types of cells is significantly impaired after saliva interaction compared to biomaterials noncontaminated with saliva. In addition, salivary pellicle formation promoted specific conditions for bacterial adhesion and proliferation.Clinically, there are no studies demonstrating that early saliva interaction is a factor for direct biomaterial rejections or infections. However, the saliva interaction can alter early biological responses at the surgical site that should be prevented. Efforts to control saliva invasion in surgical sites involving biomaterials for oral regeneration must be maximized to maintain all the basic physical–chemical–biological properties of the biomaterials.

## Figures and Tables

**Figure 1 ijms-23-02024-f001:**
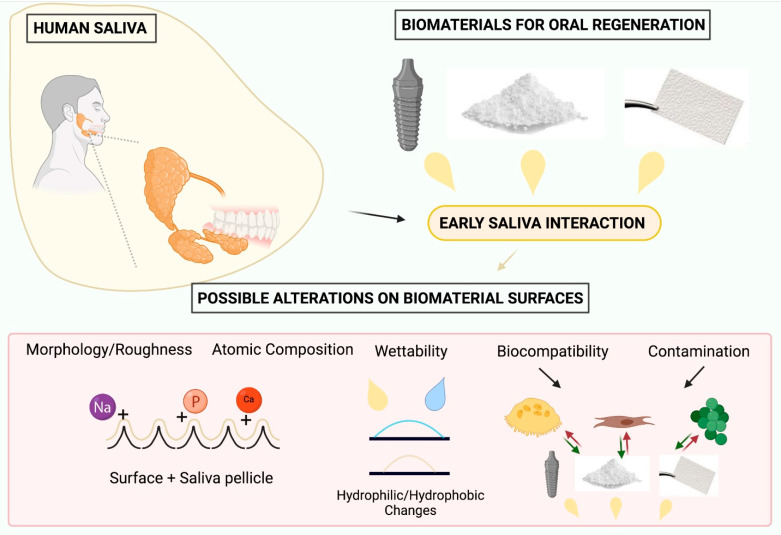
Scheme demonstrating the possible changes in biomaterials for oral regeneration after early saliva interaction. Saliva pellicle formation, wettability alterations and stimulation/repulsive responses for each type of cell were the most significant alterations. Created with BioRender.com.

**Figure 2 ijms-23-02024-f002:**
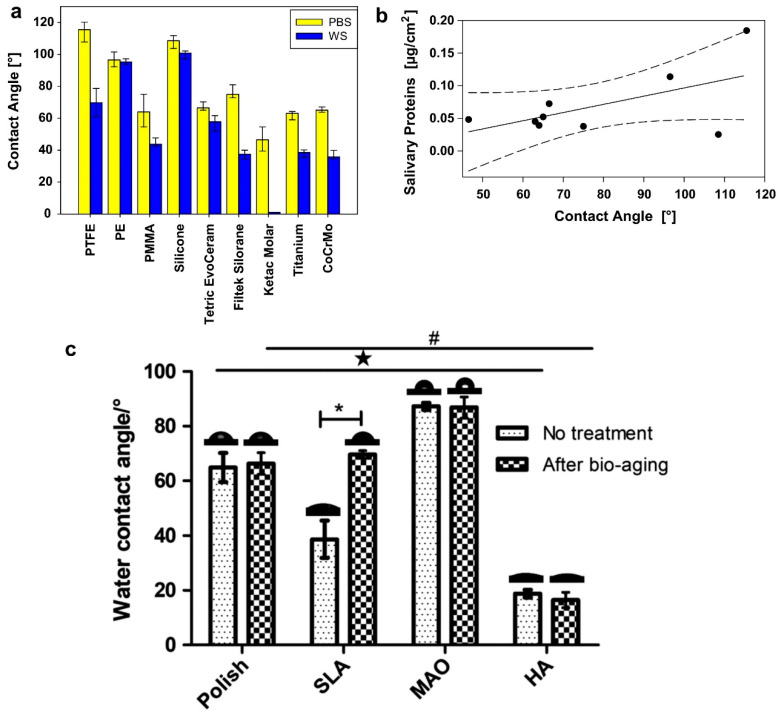
Reports about wettability characteristics and saliva protein adsorption (hydrophobic features demonstrated an increased level of saliva protein adsorption). (**a**) Different biomaterials and the corresponding contact angle after (WS) and before (PBS) saliva interaction, (**b**) followed by the level of protein adsorption for each different contact angle; images reproduced and adapted with permission from [45]. (**c**) Alterations in the measurements of contact angle after bioaging in human saliva for different surface treatments; saliva interaction affected hydrophilic properties in SLA treatment, * represent significance in the same group and # ☆ represent significance between different groups; image reproduced and adapted with permission from [47].

**Figure 3 ijms-23-02024-f003:**
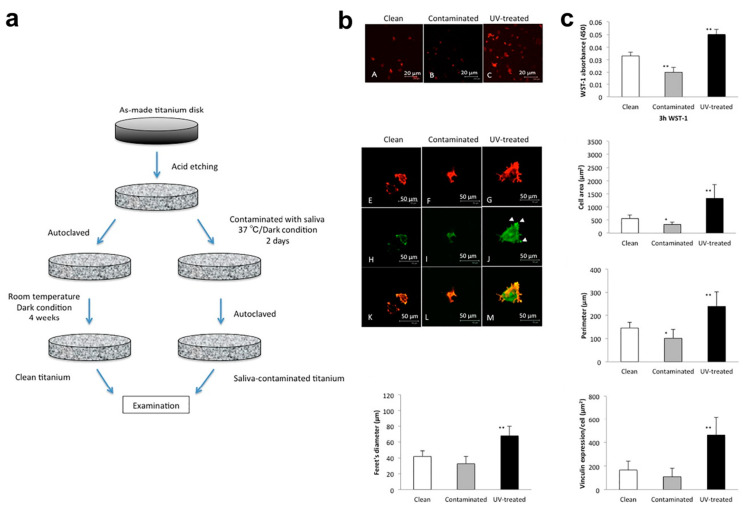
Osteoblast response after culture on saliva-contaminated surfaces for dental implants. Methodology for exposing surfaces to saliva (**a**); osteoblast morphology showing reduced spreading characteristics for contaminated surfaces (**b**); and significantly lower levels of important behavior features (Feret’s diameter, WST-1 absorbance, cell area, perimeter and Vinculin expression) for osteoblasts seeded on contaminated surfaces with saliva (**c**); * *p* < 0.05, ** *p* < 0.01. Images reproduced and adapted with permission from Elsevier, reference [21].

**Figure 4 ijms-23-02024-f004:**
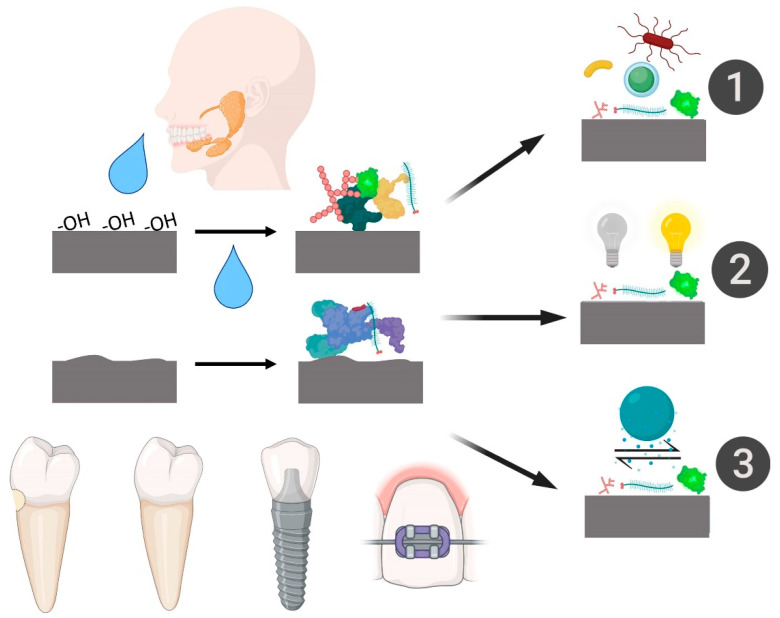
Scheme showing the diverse early effects of saliva contamination on different dental surface substrates. 1—Promotion of bacterial adhesion and colonization; 2—Promotion of surface staining; 3—Changes in the wettability, promoting different chemical interactions. Image reproduced with permission from Elsevier, reference [1].

**Figure 5 ijms-23-02024-f005:**
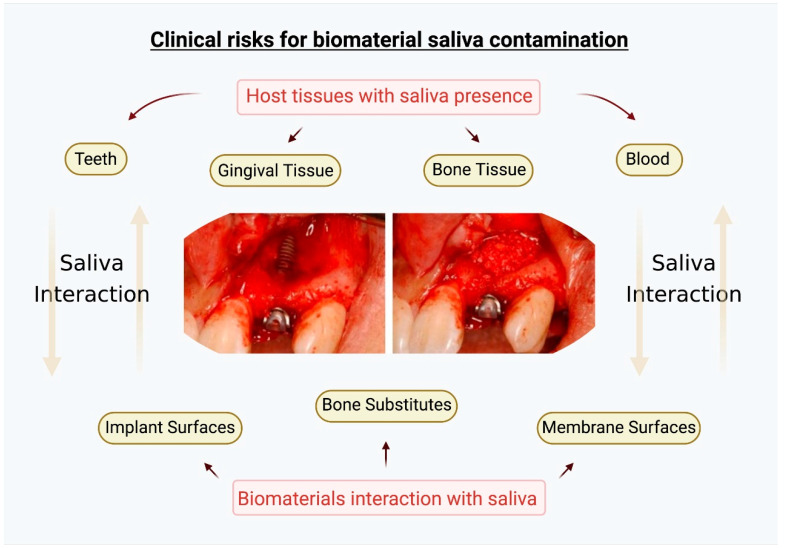
Scheme demonstrating the risks for biomaterial contamination with saliva in the oral environment when submitted to surgical procedures. Oral tissues are usually in constant contact with saliva, and the placement of biomaterials may generate early interactions. Created with BioRender.com.

**Table 1 ijms-23-02024-t001:** Main groups of proteins found in human saliva and main functions.

Protein	Main Function	Percentages
Mucin	Protection, lubrication, bolus, inhibition of demineralization	~20%
Amylase	Digestion	~25%
bPRP (basic proline-rich proteins	Lubrication and remineralization	~20%
“S” Cystatins	Protection	~8%
aPRP (acidic proline-rich proteins)	Lubrication and remineralization	~12%
gPRP (glycosylated proline-rich proteins)	Lubrication and remineralization	~5%
Immunoglobulins	Protection	~5%

**Table 2 ijms-23-02024-t002:** Studies showing saliva interaction and biocompatibility responses in dental materials for bone regeneration.

Reference	Study Model	Cells or Animals Employed	Findings	Biomaterial Applied
Zhou et al. [47]	In vitro	HGFs cell seemed over the surfaces.	Decreased adhesion and proliferation of HGF cells after bioaging in saliva.	Dental implant surfaces.
Shams et al. [52]	In vitro	MG63 human osteoblasts.	Saliva contamination altered morphology and proliferation of osteoblasts.	Dental implant surfaces.
Kunrath et al. [14]	In vitro	Osteoblast cell line MC3T3-E1.	Saliva interaction reduced the viability of osteoblast cell line.	Dental implant surfaces.
Hirota et al. [21]	In vitro	Bone marrow cells from rats.	Saliva contamination impaired osteoblastic behavior.	Dental implant surfaces.
Zöller and Zentner [53]	In vitro	Human gingival fibroblasts-like cells.	Saliva contaminated surfaces had less fibroblast adhesion and proliferation.	Dental implant surfaces.
Sun et al. [54]	In vitro	Osteoblast cell line MC3T3-E1.	Histatin-1 was added to titanium surfaces promoting spreading of osteogenic cells.	Dental implant surfaces.
Jinno et al. [56]	In vivo	Sheep.	Contaminated saliva from a human with periodontitis was interacted (15s) with the implants before insertion. Osseointegration was prejudiced regarding BIC measurements by saliva contamination.	Dental implants.
Sun et al. [60]	In vivo	Sprague–Dawley rats.	The study proposed the addition of histatin-1 (saliva protein) to absorbable collagen sponge. The results showed high bone volume when the functionalized membrane was applied.	Membranes.
Proksch et al. [57]	In vitro	Murine MC3T3 osteoblasts.	Saliva interaction hampers the osteoblast behavior. Decreased level of proliferation, alkaline phosphatase and differentiation were verified in groups with saliva.	No biomaterial applied. Cells were exposed directly to culture mediums with or without saliva.
Heaney [58]	In vitro	Human gingival fibroblasts.	Saliva interaction decreased the cell adherence to the substrate.	No biomaterial applied. Cells were exposed directly to plastic wells with or without saliva.
Pourgonabadi et al. [59]	In vitro	Bone marrow cultures and RAW 264.7 mouse macrophages.	Saliva activated polarization into proinflammatory M1 macrophages.	No biomaterial applied. Cells were exposed directly to culture mediums with or without saliva.
Mi et al. [61]	In vitro and in vivo	Human umbilical vein endothelial cells.	The study proposed the application of saliva-derived exosomes in created skin wound in mouse. The results enhanced wound healing through promotion of angiogenesis.	Wound healing.

**Table 3 ijms-23-02024-t003:** Studies showing saliva interaction with bacteria on different materials.

Reference	Study Model	Bacterial Information	Results	Biomaterial Applied
Gröbner-Schereiber et al. [80]	In vitro	*Streptococcus mutans*; *Streptococcus sanguis*	Saliva had no significant influence on the adherence of the specific strains.	Dental implant surfaces.
Mabboux et al. [70]	In vitro	*S. sanguinis*; *S. Constellatus*	Results showed that the physical–chemical properties of bacterial cells were influential on the bacterial adherence to surfaces with saliva contact.	Dental implant surfaces.
Hauser-Gerspach et al. [71]	In vitro	*S. sanguinis*	The bacterial vitality depends on the physical–chemical properties of the substrate.	Dental implant surface
Bürgers et al. [66]	In vitro	*Candida albicans*	Mucin protein serves as a receptor for *C. albicans* adherence and albumin may act as a blocker for this specific adhesion.	Dental implant surfaces.
Zhou et al. [47]	In vitro	*S. sanguinis*	Bacterial adhesion was promoted by bioaging in saliva.	Dental implant surface.
Dorkhan et al. [67]	In vitro	*S. oralis*	Saliva pellicle enhanced the bacterial metabolic activity.	Dental implant surfaces.
Dorkhan et al. [69]	In vitro	*S. oralis*	Saliva pellicle associated with rougher surfaces promoted high bacterial adherence.	Dental implant surfaces.
Cavalcanti et al. [65]	In vitro	*C. albicans*	Saliva contamination induced high virulence for *C. albicans*.	Dental implant surfaces.
Lima et al. [81]	In vitro	*S. mutans*; *Actinomyces naeslundii*	Saliva exposure did not create significant attachment of bacteria compared to noncontaminated surfaces with saliva.	Dental implant surfaces.
Li et al. [76]	In vitro	Natural saliva (wide number of microorganisms)	The substrate is significant to the proliferation of microorganisms. Biotic substrates promote rich environment for bacterial growth.	Different materials for oral regeneration (natural tissues, titanium and hydroxyapatite).
Mukai et al. [77]	Clinical	Human saliva (Wide number of microorganisms)	The study showed nonsignificance between the specificity of bacteria attached to each material. However, all materials demonstrated bacterial adhesion after contamination with saliva.	Different biomaterials for oral regeneration.
Carlen et al. [78]	In vitro	*P. gingivalis*; *F. nucleation*; *A. naeslundii*; *A. viscosuos*	The study suggested that the salivary pellicle could mediate the adhesion of bacteria present in gingivitis and periodontitis.	Hydroxyapatite beads.
Lee et al. [79]	In vitro	*E. coli* and *S. mutans*	Saliva pellicle did not promote bacterial proliferation. The material showed antibacterial properties even when saliva-coated.	Materials for oral rehabilitation (PMMA).
Turri et al. [82]	Clinical study	Biofilm oral flora; Investigation focused on *Staphylococcus spp.*	The membrane exposure to the oral cavity promoted a higher presence of bacteria compared to teeth surfaces exposed under the same conditions.	Membranes for guided oral regeneration (e-PTFE and d-PTFE).

## Data Availability

Not applicable.

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
