# Peer review of "The Impact of Early Saliva Interaction on Dental Implants and Biomaterials for Oral Regeneration: An Overview"

_ijms, 2022, doi:10.3390/ijms23042024_

Round 1

Reviewer 1 Report

I would first like to thank the Editorial Office and the authors for allowing me to read and review this article. The topic seems extremely interesting and the article well written.

Some observations:

- Lines 66-68. "It is well known that saliva has some influence on all biomaterials inserted in the oral environment, especially on those that are subjected to exposure for long periods of time" Such as? Please clarify.

- Lines 78-80. Why didn't the authors also use Scopus as a search database?

- Lines 228-230. It has been published that the use of a rubber dam during implant placement could decrease significative the amount of saliva present around the planned insertion site (doi: 10.1563/aaid-joi-D-20-00196). It could be interesting to mention and discuss it.

- In general, what type of revision was performed? Is this meant to be a narrative review? Were any specific guidelines followed? Please specify it in the title and through the text.

Author Response

Reviewer #1

Dear reviewer, we really appreciate your time and efforts to revise our manuscript. The comments were revised one by one and certainly improved the quality of our manuscript. Below, you can find the corrections comment by comment. In addition, it is highlighted in the manuscript.

1- Lines 66-68. "It is well known that saliva has some influence on all biomaterials inserted in the oral environment, especially on those that are subjected to exposure for long periods of time" Such as? Please clarify.

Thank you for your comment. The sentence was clarified with examples of biomaterials applied in oral rehabilitation with long periods of time exposed to saliva.

2- Lines 78-80. Why didn't the authors also use Scopus as a search database?

The database Scopus was added to the search, therefore, more references were added to the manuscript as can be seen highlighted in the manuscript.

3- Lines 228-230. It has been published that the use of a rubber dam during implant placement could decrease significative the amount of saliva present around the planned insertion site (doi: 10.1563/aaid-joi-D-20-00196). It could be interesting to mention and discuss it.

Thank you for the suggestion. The article was added to the reference list and discussed in the suggested sentence.

4- In general, what type of revision was performed? Is this meant to be a narrative review? Were any specific guidelines followed? Please specify it in the title and through the text.

Thank you for your comment on this subject. The manuscript is a topic overview. Chapter 2 provides all the guidelines applied to the research and methods for article selection. The title was modified as suggested by the reviewer; moreover, the word “review” was added in the main chapters of this manuscript to clarify it.

Reviewer 2 Report

The article entitled “The Impact of Early Saliva Interaction on Dental Implants and Biomaterials for Oral Regeneration” aimed to investigate an overview of all possible physical-chemical-biological changes caused by the early saliva interaction in dental implants and materials for oral regeneration. Authors have well revised several issues; however, I ask authors to add some key concepts.

- In the title it is necessary to indicate what type of review is performed by the authors

- The authors should evaluate from a microbiological point of view related findings on pathogens as P. gingivalis, which produce virulence factors that facilitate implant failure and induce dysbiotic inflammatory responses such as capsules, fimbriae, and proteases (please, see and discuss DOI 10.1155 / 2018/5326340 and DOI 10.3390 / jcm9010284)

- What are the limits of the study?

- Please, figure 2 and figure 5 are not very sharp, replace them (especially number 5)

- The conclusion should be more supported by the results.

According to this Reviewer’s consideration, novelty and quality of the paper, publication of the present manuscript is recommended after minor revision

Author Response

Reviewer #2

Dear reviewer, we really appreciate your time and efforts to revise our manuscript. The comments were revised one by one and certainly improved the quality of our manuscript. Below, you can find the corrections comment by comment. In addition, it is highlighted in the manuscript.

1- In the title it is necessary to indicate what type of review is performed by the authors

The title was corrected as suggested by the reviewer.

2- The authors should evaluate from a microbiological point of view related findings on pathogens as P. gingivalis, which produce virulence factors that facilitate implant failure and induce dysbiotic inflammatory responses such as capsules, fimbriae, and proteases (please, see and discuss DOI 10.1155 / 2018/5326340 and DOI 10.3390 / jcm9010284)

Thank you for the suggested references. The articles were cited and discussed in chapter 5.2.1.

3- What are the limits of the study?

We appreciate this comment. A new paragraph was added to chapter 7 (last paragraph) with the limitations of this review.

4- Please, figure 2 and figure 5 are not very sharp, replace them (especially number 5)

Thank you for your comment. Figure 2 is a reprint from other articles, the figures are reprinted exactly equal to the quality published in the original articles, and we can’t interfere with that aspect. The information provided by the figures is important to support the chapter showing the alterations on the wettability.

Figure 5 was replaced by a figure with high quality. However, the information provided by the figure is demonstrated in a schematic view, only using the clinical image to correlate to the clinic procedures.

5- The conclusion should be more supported by the results.

The conclusion section was corrected to clarify and to provide more information correlated with the studies investigated in this review.